# An Exosomal Urinary miRNA Signature for Early Diagnosis of Renal Fibrosis in Lupus Nephritis

**DOI:** 10.3390/cells8080773

**Published:** 2019-07-25

**Authors:** Cristina Solé, Teresa Moliné, Marta Vidal, Josep Ordi-Ros, Josefina Cortés-Hernández

**Affiliations:** 1Hospital Universitari Vall d’Hebron, Vall d’Hebron Research Institute (VHIR), Lupus Unit, 08035 Barcelona, Spain; 2Hospital Universitari Vall d’Hebron, Department of Renal Pathology, 08035 Barcelona, Spain

**Keywords:** lupus nephritis, renal fibrosis, urinary exosomes, multi-marker panel

## Abstract

For lupus nephritis (LN) management, it is very important to detect fibrosis at an early stage. Urinary exosomal miRNAs profiling can be used as a potential multi-marker phenotyping tool to identify early fibrosis. We isolated and characterised urinary exosomes and cellular pellets from patients with biopsy-proven LN (*n* = 45) and healthy controls (*n* = 20). LN chronicity index (CI) correlated with urinary exosomal miR-21, miR-150, and miR-29c (r = 0.565, 0.840, −0.559, respectively). This miRNA profile distinguished low CI from moderate-high CI in LN patients with a high sensitivity and specificity (94.4% and 99.8%). Furthermore, this multimarker panel predicted an increased risk of progression to end-stage renal disease (ESRD). Pathway analysis identified *VEGFA* and *SP1* as common target genes for the three miRNAs. Immunohistochemistry in LN renal biopsies revealed a significant increase of COL1A1 and COL4A1 correlated with renal chronicity. SP1 decreased significantly in the high-CI group (*p* = 0.002). VEGFA levels showed no differences. In vitro experiments suggest that these miRNA combinations promote renal fibrosis by increasing profibrotic molecules through SP1 and Smad3/TGFβ pathways. In conclusion, a urinary exosomal multimarker panel composed of miR-21, miR-150, and miR-29c provides a non-invasive method to detect early renal fibrosis and predict disease progression in LN.

## 1. Introduction

Fibrosis, or tissue scarring, is the result of an excessive, persistent, and destructive accumulation of extracellular matrix components (ECM) in response to chronic tissue injury in the kidney. Renal fibrosis represents the final stage of most chronic kidney diseases and contributes to a progressive and irreversible decline in kidney function. The continuous accumulation of ECM causes a disruption of the epithelium and/or endothelium integrity that results in the activation of a complex cascade of molecular and cellular events [1]. Initially, there is an inflammatory response with the release of profibrotic cytokines, chemokines, and growth factors, which in turn promote the proliferative phase of the scarring process characterised by the recruitment and activation of fibroblasts into ECM-secreting myofibroblasts and the formation of a permanent fibrotic scar [2,3].

Lupus nephritis (LN) occurs in ~40–75% of systemic lupus erythematosus (SLE) patients [4] and continues to be one of the major causes of morbi-mortality in those patients. Despite improvements in its management, up to 20% of patients will progress to end-stage renal disease (ESRD) [5]. So far, renal biopsy continues to be the gold-standard technique to evaluate the degree of fibrosis, but its invasiveness makes it unsuitable for serial monitoring [6]. Detection of early stages of fibrosis could contribute to identify patients at risk of progressing to ESRD who can benefit from new therapeutic approaches. Therefore, there is the need to identify non-invasive biomarkers to detect early fibrosis and to monitor its progression.

Urine samples are obtained relatively easily and cost-efficiently compared with other biological samples such as blood, cerebrospinal fluid, or tissue biopsy. The analysis of urinary miRNAs provides a measure of the health of the excretory system and is detected noninvasively from urinary cellular pellet or inside urinary exosomes [7]. Most studies use urinary cellular pellet for study of miRNA expression [8,9,10]. In recent years, urinary exosomes gathered strength in the field of biomarkers due to being small membranous vesicles secreted by a variety of living cells and implicated in cell-to-cell communication [11,12]. Inside them, there are small non-coding RNAs, the miRNAs, which regulate many molecular and cellular processes by repressing translation or causing breakdown in target gene mRNAs [13]. Urinary exosomal miRNAs can accurately reflect renal dysfunction and structural damage, making them good targets for the exploration of biomarkers for urinary tract diseases [14,15].

miRNA expression profiling in renal fibrosis has been evaluated in kidney tissue samples [16], blood [17], and urine [18] from various chronic kidney diseases. Despite a large number of miRNAs being identified, a recent meta-analysis study only showed seven of them to be highly related to fibrosis [19]. Of those, miR-29c, miR-21, and miR-200a, regulators of the TGF-β/Smad3 signalling pathway, have been described in urinary exosomes [20,21,22,23]. Furthermore, three miRNAs have been associated with renal fibrosis in LN patients: miR-410, which contributes to renal fibrosis by inhibiting interleukin-6 [24], miR-29c as predictor of early renal fibrosis [25], and miR-150, which promotes renal fibrosis by downregulation of SOCS1 [26].

Unfortunately, it is now generally accepted that single markers do not achieve sufficient sensitivity and specificity for diagnosis and routine clinical practices. A novel trend based on the combination of several signatures has improved clinical performance [27,28,29]. In this study, we compared miR-29c, miR-200a, miR-21, miR-410, and miR-150 expression levels between urinary pellet and exosomes. Then, we estimated a combinatory panel of urinary exosome-derived miRNAs from active LN patients to identify early fibrosis and predict progression to ESRD.

## 2. Materials and Methods

### 2.1. Patients and Samples

Patients with biopsy-proven active LN were recruited from the Lupus Unit at Vall d’Hebron Hospital (*N* = 45). The study was ethically approved by Vall d’Hebron Hospital Ethics Committee and all patients provided written informed consent prior inclusion. All patients fulfilled at least 4 of the American College of rheumatology (ACR) revised classification criteria for SLE [30]. Healthy donors were used as controls (*N* = 20). Repeated renal biopsy was performed if required according to clinical protocol (*N* = 7). Urine samples were collected from each patient 1 day before renal biopsy and processed immediately to be stored at −80 °C (more details in Appendix A). SLE disease activity was assessed by the SLE Disease Activity Index 2000 update (SLEDAI-2Ks; range 0–105) [31]. Patients were classified according to the chronicity index (CI) at renal biopsy: Low CI (<2, *N* = 18), moderate CI (2–4, *N* = 21), and high CI (≥4, *N* = 6). ESRD progression was defined by an estimated glomerular filtration rate (eGFR) <15 mL/min/1.73 m^2^, the initiation of renal replacement therapy or receiving kidney transplantation, or 40% reduction of baseline eGFR [32]. Estimated glomerular filtration ratio (eGFR) was calculated using the chronic kidney disease epidemiology collaboration (CKD-EPI) formula [33].

### 2.2. Renal Histology and Immunohistochemistry

Renal biopsies were examined by light and immunofluorescence microscopy following standard methods and categorised according to the International Society of Nephrology/Renal Pathology Society classification (2003) [34] and rated for activity (AI) and chronicity (CI) [35]. Renal chronicity was measured by Gömöri trichrome staining from paraffin-embedded renal samples. For immunohistochemistry, rabbit anti-VEGF-A antibody (Abcam, ab46154, dilution of 1:50) and a diaminobenzidine (DAB) chromogen kit (DAKO) were used. Histology and immunohistochemistry samples were blinded and scored by the Vall d’Hebron pathologist unit (more details in Appendix A).

### 2.3. Exosome and Pellet Purification and Characterisation

Fresh urine samples (50 mL) to obtain the urinary cellular pellet and supernatant were centrifuged at 3900× *g* for 30 min and kept for urinary exosome isolation. Following the manufacturer’s instructions, urinary exosomes were isolated using the miRCURY™ Exosome Isolation Kit—Cells, urine and CSF (Exiqon, Woburn, Massachusetts, USA). In order to validate the exosome purification protocol, urinary exosome morphology, shape, and size were analysed by Cryo-transmission electron microscopy. NanoSight and Western blot analysis were used to characterise exosome isolation (more details in Appendix A). Quantification of Urinary exososmes was performed byFluorCet Exosome Quantitation Kit (SBI, more details in Appendix A) according to the manufacturer’s instructions.

### 2.4. RNA Isolation from Urinary Cell Pellet and Exosomes

RNA from urinary pellets or exosomes were extracted using the miRCURY™ RNA Isolation Kit-Cell & Plant (Exiqon, Woburn, MA, USA, more details in Appendix A), following the manufacturer’s instructions. Quantification and evaluation of the RNA quality was assessed by Bioanalyzer PicoChip analysis.

### 2.5. Individual Assay qPCR-RT

Initially a first-strand cDNA synthesis reaction was made to provide a template for all microRNA real-time PCR assay using the miRCURY LNA™ Universal RT microRNA PCR (Exiqon, Woburn, MA, USA, more details in Appendix A). MiRNAs were quantified with specific miRCURY LNA primer set and ExiLENT SYBR Green master mix (Exiqon, Appendix A) using the ABI PRISM 7000 following the manufacturer’s instructions. Data were normalised based on the expression of RNU6 and relative quantification was calculated (RQ = 2^−ΔΔCt^) using the Livak method [36].

### 2.6. Pathway Analysis

To gain insight into the functions of miRNA target genes, we performed gene ontology (GO) classification and Kyoto Encyclopedia of Genes and Genomes (KEGG) pathway enrichment analysis using the online tools of DNA ntelligent Analysis (DIANA)-miRPath v3.0 software [37]. To present the regulation between miRNA and gene, experimentally validated targets were extracted for identified miRNAs from miRTarBase [38] and critical miRNA–target interactions were constructed using miRNet (www.mirnet.ca) [39].

### 2.7. Immunofluorescence in Renal Biopsy

Immunofluorescence was performed on paraffin-embedded (FFPE) renal biopsies during renal flare (*N* = 3 for each subgroup) following the protocol described by Mason et al. [40]. Staining was performed with 1:50 rabbit anti-SP1 (Abcam, ab124804), 1:100 mouse anti-COL1A1 (Abcam, ab6308), or 1:100 mouse anti-COL4A1 (Abcam, ab6311) overnight at 4 °C (more details in Appendix A).

### 2.8. Human Kidney Cells Culture

Primary human renal mesangial cells (RMCs) and renal tubular epithelial cells (RTCs) were purchased from InnoProt (Derio, Spain) and cultured in the recommended media provided by the manufacturer. Cells were grown at 37 °C in a humidified 5% CO_2_ atmosphere. Cell passes were performed using TrypLE™ Express (Gibco^®^ ThermoFisher Scientific, Waltham, MA, USA).

### 2.9. Overexpression of miR-21/miR-150 and Inhibition of mir-29c in Human Kidney Cells

Cells plated on 24-well plates were transfected with mimic miR-21 and mimic miR-150 (ThermoFisher Scientific, Waltham, MA, USA) or with miR-29c inhibitor (ThermoFisher Scientific, Waltham, MA, USA) using Lipofectamine RNAiMAX (Invitrogen, Carlsbad, CA, USA) following the manufacturer’s instructions. After 48 h, cells were stimulated with TGFβ1 citokine (10 ng/mL, ThermoFisher Scientific, Waltham, MA, USA). After 24 h, the total RNA was extracted using miRCURY RNA Isolation Kit (Exiqon, Woburn, MA, USA) or the immunofluorescence was performed (more details in Appendix A).

### 2.10. Luciferase Activity Analysis

Human kidney cells were plated in 96-well plates for 24 h and then were cotransfected using DharmaFECT Duo transfect reagent (ThermoFisher Scientific, Waltham, MA, USA) with 40 ng of pEZX-MT06-SP1 3′UTR plus 10 nM of miR-21, miR-150, or miR-29c analog or miR negative control (5 duplicates per group). The pEZX-MT06-SP1 3′UTR plasmid contains firefly luciferase and miR binding site from SP1 3′UTR, which is inserted between firefly luciferase cDNA and PolyA (more binding site details in Appendix A). This plasmid also includes Renilla luciferase as an internal control (GeneCopoeia, Rockville, MD, USA). Transfected cells were lysed by reporter lysis buffer (Promega, Madison, WI, USA). Dual luciferase activity was measured with the Dual-Luciferase Reporter Assay System (Promega, Madison, WI, USA) by FLUOstar Omega (BMG Labtech, Oternberg, Germany).

### 2.11. Statistical Analysis

Statistical analyses were performed by GraphPad Prism version 6 (GraphPad Software, La Jolla, CA, USA) and SAS version 9.3 (SAS Institute Inc., Cary, NC, USA). The mean expression of miRNA levels was compared using Mann–Whitney U/Kruskal–Wallis H tests, as appropriate. The relationship between miRNA expression and histological/clinical parameters of LN patients were analysed using the Spearman correlation coefficient. Risk of progression to ESRD and renal survival rate across urinary exosomal miRNAs were analysed and compared using the Kaplan–Meier analysis and the log-rank test. The diagnostic performance of biomarkers was evaluated by calculating their sensitivity and specificity using receiver operating characteristic (ROC) curves. A combinatorial analysis of multiple biomarker signatures was carried out using the CombiROC method [41] (more details in Appendix A).

## 3. Results

### 3.1. Patients

Patient demographic characteristics and laboratory measurements are summarised in Table 1. Most patients included had a type IV GMN and had a moderate chronicity index degree (*N* = 21) at inclusion (Appendix A). No significant clinical differences were observed between subgroups such as age, gender, serum creatinine, proteinuria, blood urea nitrogen (BUN), and eGFR (Table 1). Patients were followed up for a median of 8 years (range: 3.5 to 12.5) after renal biopsy.

### 3.2. Isolation and Characterization of Urinary Exosomes and Cellular Pellet

Urinary exosomes and pellets were isolated as described in the methodology. Exosomes were examined using cryo-transmission electron microscopy, nanoparticle tracking, and Western blot analysis. Results revealed vesicles of 85.5 ± 33.4 nm diameter with the characteristic cup-shaped exosome morphology and the presence of the urinary exosome TSG101 protein (Figure 1A). The number of urinary exosomes was similar among the different LN groups (mean 2.3 × 10^7^, Appendix A). The urinary cellular pellet was confirmed by the negativity of TSG101 protein, confirming the absence of contamination from other cellular compartments.

### 3.3. miR-29c, miR-200a, miR-21, miR-410, and miR-150 Expression Levels Measured by Quantitative Reverse Transcription-Polymerase Chain Reaction (qPCR-RT) in Urinary Exosomal Preparations and Cellular Pellet

A deep sequencing analysis was performed to compare the miRNA composition in the urinary exosomes and cellular pellets obtained from three patients with LN (Appendix A). The highest percentage of miRNA was extracted from exosomes compared with the cell pellet (total reads 41,994,100 vs. 15,545,900, Appendix A), suggesting that there is an enrichment of miRNA in exosomes. Next, we compared the expression levels of the five miRNAs (miR-29c, miR-200a, miR-21, miR-410, and miR-150) in urinary exosomes and cell pellet using qPCR-RT. Expression levels of the study miRNAs were only detected in urinary exosomes (Appendix A). For these reasons, we decided to focus on urinary exosomes as the source to identify miRNA biomarkers for LN renal fibrosis.

We found miR-21 and miR-150 to be significantly overexpressed in the LN group compared with healthy controls (6.6- and 2.3-fold change, respectively) and miR-410 and miR-29c to be significantly reduced (−1.8- and −2.2-fold change, respectively, Figure 1B). No significant differences were observed in miR-200a-3p expression levels (Appendix A). When results were evaluated according to the CI subgroups, miR-410 expression levels did not differ between subgroups (Appendix A). However, miR-21-5p, miR-150, and miR-29c were strongly correlated with renal chronicity (r = 0.565, r = 0.840, and r = −0.559, respectively, Appendix A). Both miR-21 and miR-150 expression levels increased progressively according to the degree of CI, being most highly expressed in the high CI group (13.3- and 4.8-fold change, respectively). However, miR-29c showed a progressive downregulation, being the lowest expression levels found in the high CI group (−18.7-fold change, Figure 1C). No correlations were found between miR-21, miR-150, and miR-29c with traditional clinical parameters of damage progression, such as serum creatinine levels, proteinuria or eGFR, and the activity index score. However, significant correlation was found with tubular atrophy, interstitial fibrosis, glomerular sclerosis, and fibrous crescent (*p* < 0.05, Appendix A).

### 3.4. CombiROC Performance Analyses for Optimal Marker Combinations

We next tested miR-21, miR-150, and miR-29c prognostic value for the early detection of renal fibrosis in LN. The diagnostic/prognostic accuracy of individual and of multiple marker combinations was calculated by uploading the original data of analyte concentrations to the CombiROC tool. A test cut off value of 1.80 (control mean + 3SD) was used. All possible marker combinations to distinguish low CI from moderate-high CI group were obtained separately by setting the minimum sensibility and specificity values in the next step and finally the best performing individual markers and combinations were obtained via ROC analysis (Figure 2A,B).

As an individual biomarker, miR-150 had the best profile to distinguish the two CI groups (AUC = 0.970, cut-off > 0.903 with 96% sensibility and 83% specificity) whereas miR-21 ROC curve was the least predictive (AUC = 0.742, cut-off > 4.15 with 81% sensibility and 72% specificity). ROC analysis of other clinical parameters such as creatinine, eGFR, or proteinuria levels as predictors of early renal fibrosis did not show better predictive value (Figure 2B).

Interestingly, the multimarker panel of miR-29c/mir-150/miR-21 expression levels exhibited the highest sensitivity and specificity values (94.4% and 99.8%, AUC = 0.996, respectively, Figure 2B). The pie chart of three miRNAs multimarker panel shows a low percentage of false positive cases for this model (Appendix A).

We further investigated whether the combinatory multimarker panel could predict disease progression in LN. Kaplan–Meier survival analysis showed that multimarker positive LN patients (miR-29c/miR-150/miR-21) had 5- and 10-year renal survival rates of 73.6% and 63.2%, respectively, in contrast with 100% and 89.4% for negative patients. The difference in renal survival rate was statistically significant according to log-rank test (Figure 2C, *p* = 0.027).

Seven patients had a repeated biopsy in a median time of 23.8 months (range from 12 to 96 months) (Appendix A). We observed a significant increase in miR-21 and miR-150 expression levels (2.7- and 3.1-fold change, respectively) and miR-29c reduction (−6.2-fold change, *p* < 0.05, Figure 2D) between baseline and repeated samples that correlated with a CI increase.

### 3.5. Pathway Enrichment Analyses

We next hypothesised the biological pathways those selected miRNAs may regulate by targeting multiple pathway-specific mRNAs. For each miRNA, we retrieved validated targets from miRTarBase, (Appendix A). An interaction network was constructed using miRNet (Figure 3A). Only two mRNAs were common to the three miRNAs, *SP1* and *VEGFA*. Since LN patients with renal fibrosis overexpressed miR-150 and miR-21 but downregulated miR-29c, we also performed GO and KEGG pathway enrichment analysis using miRPath (*p* value < 0.01, Figure 3B). Significant enrichment of target genes was detected for seven functional GO pathways associated with extracellular matrix and collagen formation (Figure 3B). In KEGG pathway analysis, three pathways were detected where ECM-receptor interaction was the most significant (Figure 3B). The predicted interactions of this pathway showed that *COL1A1* and *COL4A1* were one of the most significant modified mRNA genes common for the three miRNAs (Appendix A).

### 3.6. Confirmation of SP1 as Common Urinary miRNAs Target and Associated Profibrotic Molecules in Kidney Biopsies

Next, we sought to confirm the pathway analysis in the kidney biopsies from LN patients. Therefore, we examined the protein levels of SP1 and VEGFA in the different CI subgroups. Immunohistochemical staining of VEGFA in the glomeruli was similar in all LN groups (Appendix A). However, high tubular staining of SP1 protein was observed in low and moderate CI patients but SP1 decreased significantly in high CI patients (*p* = 0.004 and 0.002, respectively, Figure 3C). COL1A1 and COL4A1 proteins were predominantly localised in the tubular epithelial cells and extracellular matrix, increasing the staining progressively with the degree of chronicity (Figure 3C).

To confirm *SP1* as a target for the three studied miRNAs, we performed luciferase assay studies using primary human renal mesangial cells (RMCs) and renal tubular epithelial cells (RTCs). We co-transfected RMCs and RTCs with a plasmid containing a luciferase gene under the control of SP1 3′ untranslated region (UTR) and with either miR-29c, miR-150, or miR-21 analogue. MiR analogue negative was used as control. Luciferase activity decreased in RMCs by 41.2%, 63.9%, and 54.9% 48 h after the transfection in the presence of miR-29c, mir-150, and miR-21 analogue compared with the negative control, respectively (Figure 3D). In RTCs, the luciferase activity reduction was more pronounced in all the three (68.8%, 66.8%, and 67.9%, respectively, Figure 3D).

### 3.7. Over-Expression of miR-21-5p/miR-150 and Inhibition of miR-29c Increase Profibrotic Proteins

In order to better investigate the roles of miR-21/miR-150/miR-29c in LN renal fibrosis formation, we overexpressed both miR-21/miR-150 and inhibited miR-29c in TGFβ-stimulated RMCs and RTCs. Overexpression of miR-21/miR-150 under TGFβ-stimulated RMCs induced a significant reduction of *SP1* expression (−3.8-fold change) whereas no changes in expression levels were observed with miR-29c inhibition in this cell type (Figure 4A). The overexpression of miR-21/Mir-150 or the inhibition of miR-29c induced *COL4A1* formation via Smad3/TGFβ pathway (3.8- and 4.6-fold change, respectively, Figure 4A).

However, overexpression of *SP1* was only observed in miR-29c-inhibited RTCs along with an increased expression of the study profibrotic molecules. Overexpression of miR-21/miR-150 in RTCs induced a significant reduction in *SP1* expression levels, whereas *COL1A1*, *COL4A1*, *TGFβ*, and *Smad3* were significantly increased suggesting a Smad3/TGFβ fibrosis-dependant pathway, independent of SP1.

Immunofluorescence staining of SP1 was significantly increased in miR-29c inhibited RTCs (*p* = 0.003, Figure 4B). In contrast, SP1 was inhibited in miR-21/mir-150 transfected renal cells (Figure 4B). COL1A1 protein levels increased in all conditions and did not correlate with SP1 levels (Figure 4B).

## 4. Discussion

We have shown that urinary exosomes are the best source of urinary miRNA biomarkers in LN patients. The comparison between urinary pellet and exosomes showed a better yield of miRNAs from exosomes than from the cellular pellet. One possible explanation is the fact that inside urinary exosomes, miRNAs can be preserved and protected from the RNase activity in the kidney urinary tract [42].

We simultaneously measured five urinary exosomal MiRNAs, previously associated with fibrosis: miR-21 and miR-200a, which directly regulate multiple collagen isoforms and extracellular matrix components via TGFβ pathway [23]; and miR-410, miR-29c, and miR-150, which have been reported in renal fibrosis in LN [24,25,26]. Of those, we only found miR-21, miR-150, and miR-29 to be differentially expressed in urinary exosomes from patients with LN and to correlate with chronicity scores in the renal biopsy. High degree of fibrosis was characterised by a significant upregulation of miR-21, miR-150, and miR-29c downregulation.

FibromiRs are essential downstream components of both fibrogenic and fibrosis-suppressive signalling pathways, and changes in their expression directly affect the biological response following activation of these pathways [3]. As a consequence, several reports studied them separately, as individual miRNAs assay. However, fibrogenesis results from alterations or imbalances in multiple interconnected molecular pathways [43]. We report here, for the first time, a multimarker urinary exosomal miRNA panel to diagnose the degree of chronicity in lupus nephritis patients at flare using the CombiROC tool. This methodology is available as an easy-to-use web application to accurately determine optimal combinations of markers and takes advantage of combinatorial analysis and ROC curves [41]. It has been used to screen multimarker signatures for autoimmune hepatitis [44], biomarker panels for cancer [45], and Parkinson disease [46]. In our study, the individual miRNAs discriminate low from moderate-high chronicity in lupus nephritis; however, their combination as multiple biomarker signatures showed the best specificity and sensitivity profile to detect early fibrosis. In addition, our results also showed the presence of the miR-29c/miR-150/miR-21 combinatory panel as a predictor of progression to ESRD and its predictive value was superior to routine conventional biomarkers such as creatinine and eGFR. If these data are confirmed in larger studies, this biomarker panel could be used as a diagnostic marker of renal fibrosis and a prognostic biomarker of progression to ESRD.

The exact mechanism of fibrosis in LN is unknown, but the persistence of wound-healing processes with prolonged production of growth factors, fibrogenic cytokines, and proteolytic enzymes, leading to increased synthesis and degradation of the extracellular matrix [47] seems to play a role. Previous studies in LN focused on the fibrotic role of miR-150 by targeting SOCS1, a negative regulator of profibrotic proteins [26] and miR-29c downregulation [25]. Upregulation of miR-21 also induces collagen and other extracellular matrix components via the TGFβ/Smad3 pathway [48]. Our pathway analysis identified *VEGFA* and *SP1* as common targets of miR-21, miR-150, and miR-29c. The expression of VEGFA, SP1, and profibrotic proteins was then measured in the LN kidney biopsies. As expected, COL1A1 and COL4A1 staining increased progressively and correlated with the renal chronicity index. This is not surprising, since renal interstitial fibrosis is the result of an increase in important ECM components, such as collagens [49]. Although it is well-known that down regulation of VEGF is related with worse prognosis in lupus nephritis [50], we did not find differences in the VEGFA staining in the CI subgroups. However, SP1 was found to be increased in patients with low and mainly moderate CI, but significantly decreased in patients with high CI.

We next evaluated in vitro the physiologic relevance of the study miRNAs in promoting fibrosis. Transfection of PTCs with miR-150 and miR-21 mimics, significantly suppressed SP1 production while inducing an increase of profibrotic molecules after TGFβ stimulation. The same condition in mesangial cells mirrored the results obtained in the tubular cells suggesting a profibrotic effect of these miRNAs through the Smad3/TGFβ pathway, independent of SP1. Transfection of PTCs with a miR-29c inhibitor significantly increased the production of SP1, profibrotic molecules, and Smad3. In mesangial cells, there was a moderate production of COL4A1, TGFβ, and Smad3, whereas SP1 and COL1A1 were not significantly expressed. Data suggest a role for miR-29c in the production of fibrosis in tubular epithelial cells through the SP1/Smad3/TGFβ pathway.

Growing evidence demonstrates that SP1 plays an important regulatory role for the expression of several genes relevant to fibrosis, including TGFβ, vascular endothelial growth factor, type I collagen, and downstream targets of TGFβ, such as matrix metalloproteinases, plasminogen activator inhibitor-1, and fibronectin [51]. Several studies have shown that SP1 mediates TGFβ fibrogenic factor through cooperation with Smads proteins and is an essential mediator for the production of type I collagen induced by miR-29c downregulation [52]. SP1 has been shown in the glomeruli and proximal tubules of a broad spectrum of human glomerulonephritides, being more prominent in the glomeruli of secondary proliferative GNs [51]. It has been shown that SP1 mediates MCP-1 expression in murine podocytes and could also be implicated in the induction of renal injury through the attraction of macrophages [53]. Therefore, Sp1 overexpression in glomeruli of proliferative GNs may be as a result of the inflammatory process.

Our data suggest that miR-150, miR-21, and miR-29c play an important role in renal fibrogenesis by increasing the synthesis of profibrotic molecules through several mechanisms. We demonstrated an early role of miR-29c and SP1 in regulating collagen production in tubular epithelial cells, while miR-150 and miR-21 expression contribute to the maintenance and amplification of the fibrotic process leading to ESRD through a Smad3/TGFβ pathway independent of SP1 at later stages (Figure 4C).

In summary, we have demonstrated that a urinary exosomal multimarker panel formed by miR-21, miR-150, and miR-29c provide a non-invasive method to detect early renal fibrosis and predict disease progression in LN patients. Although the exact molecular basis of the three urinary exosomal miRNAs during renal injury remains unclear, we have identified relevant pathways in the LN renal fibrosis mechanism, and a better understanding of the pathogenesis could contribute to develop new therapeutic approaches.

## Figures and Tables

**Figure 1 cells-08-00773-f001:**
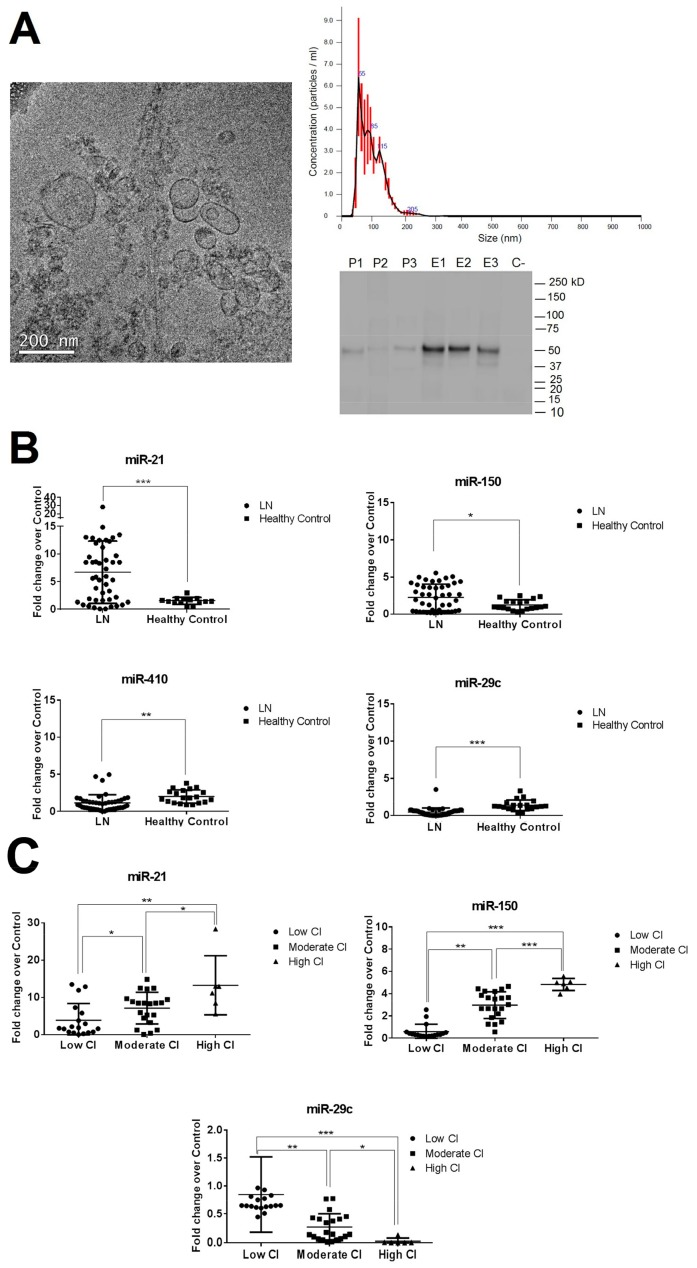
Characterisation of urinary exosomes and expression levels of urinary exosomal miR-21, miR-150, miR-410, and miR-29c in lupus nephritis (LN) patients’ urinary pellets. (**A**) Cryo-transmission electron micrograph, nanoparticle tracking analysis and Western blotting of TSG101 protein confirmed that the isolated small vesicles are urinary exosomes (size of 52–119 nm in diameter). P1–P3: Pellet urinary samples. E1–E3: Exosome urinary samples. C-: Negative control. (**B**) Values of miRNAs expression from healthy controls and LN patients at flare time were represented as individual dots. (**C**) Values of miRNAs expression when LN patients were classified according to the chronicity index at renal biopsy (low CI (<2, *N* = 18), moderate CI (2–4, *N* = 21), high CI (≥4, *N* = 6)). Values were normalised using RNU6 and fold change was calculated over healthy control group. * *p* < 0.05, ** *p* < 0.005, and *** *p* < 0.0005.

**Figure 2 cells-08-00773-f002:**
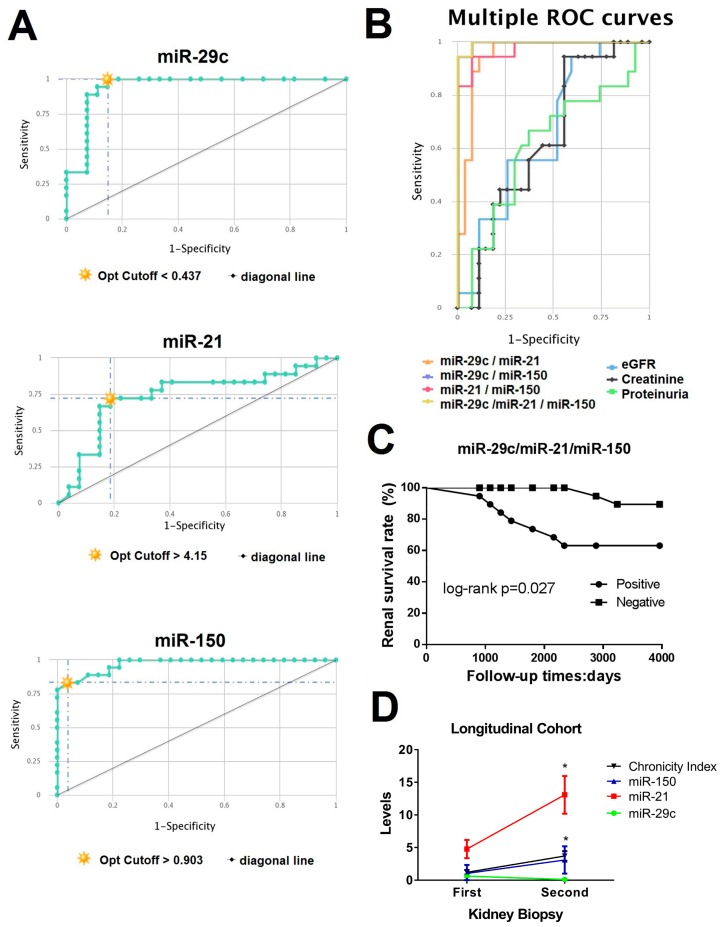
Correlation of urinary miRNAs with chronicity index and risk of end-stage progression disease (ESRD). (**A**) Receiver operating characteristic (ROC) curve analysis of individual urinary exosomal miR-29c, miR-21, and miR-150 expression to distinguish low CI group from moderate-high CI group. Opt cutoff: Optimal cutoff according Youden’s index. (**B**) Multiple ROC curves showed that miR-29c/miR-21/miR-150 multimarker panel has the best ROC curve profile. They were obtained using CombiROC online program. eGFR: Estimated glomerula filtration rate. (**C**) Kaplan–Meier curve of renal survival rate stratified by urinary exosomal miRNAs multimarker panel. There were significant differences of the renal survival rate between positive and negative patients to multimarker panel by the long-rank test. (**D**) Relative expression levels of miR-150, miR-21, and miR-29c were evaluated in seven patients with two kidney renal biopsies. Values were normalised using RNU6. * *p* < 0.05.

**Figure 3 cells-08-00773-f003:**
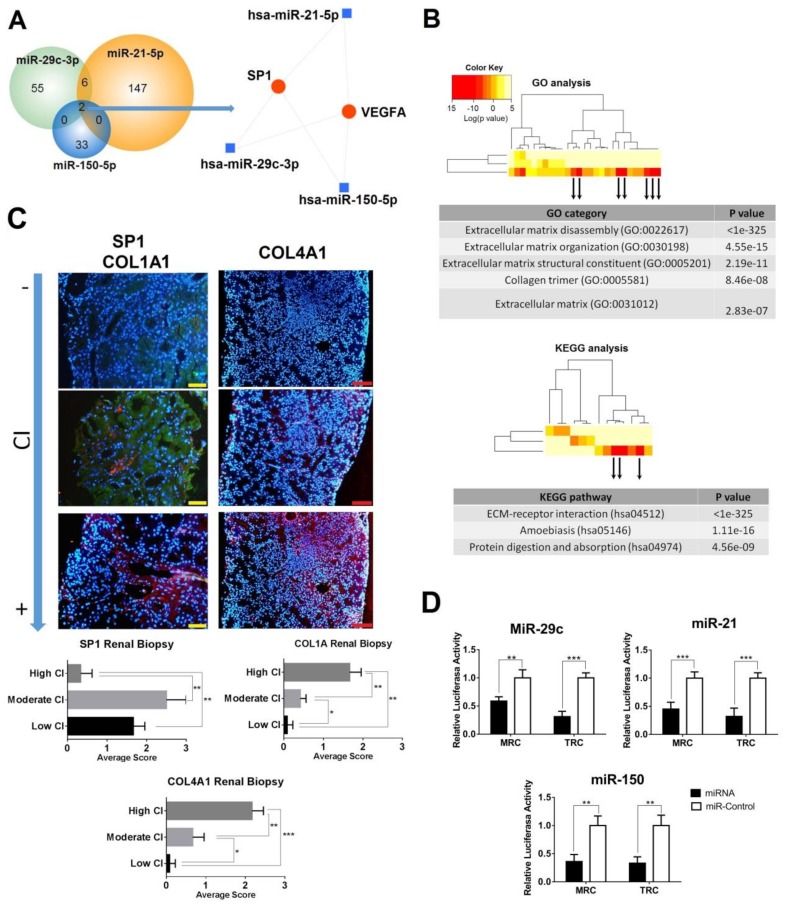
Role of urinary exosomal miRNAs in LN renal fibrosis formation. (**A**) Venn diagram representing overlap of validated targets of miR-29c-3p, miR-150-5p, and miR-21-5p. Network of miRNA–target interaction was obtained using miRNet tool. (**B**) Enriched gene ontology (GO) and Kyoto Encyclopedia of Genes and Genomes (KEGG) pathways using DIANA miRPatch v3.0 software. We focus in pathways with downregulation of miR-29c and upregulation of miR-150 and miR-21. GO: Gene Ontology. KEGG: Kyoto Encyclopedia of Genes and Genomes. CI: chronicity index. (**C**) Immunofluorescence of SP1 (green), COL1A1 and COL4A1 (red) in renal biopsy from LN chronicity subgroups (low CI, moderate CI, and high CI). DAPI staining was used to label cell nuclei. Scale bar = 100 µm. DAPI: 4’,6-diamidino-2-phenylindole. (**D**) Cotransfection of miR-21, miR-150, or Mir-29c and SP1 3’ UTR luciferase reporter significantly decreases the luciferase activity compared with miR-control (five replicates per group). MRC: Renal mesangial cells; TRC: Renal tubular cells. * *p* < 0.05, ** *p* < 0.005, and *** *p* < 0.0005.

**Figure 4 cells-08-00773-f004:**
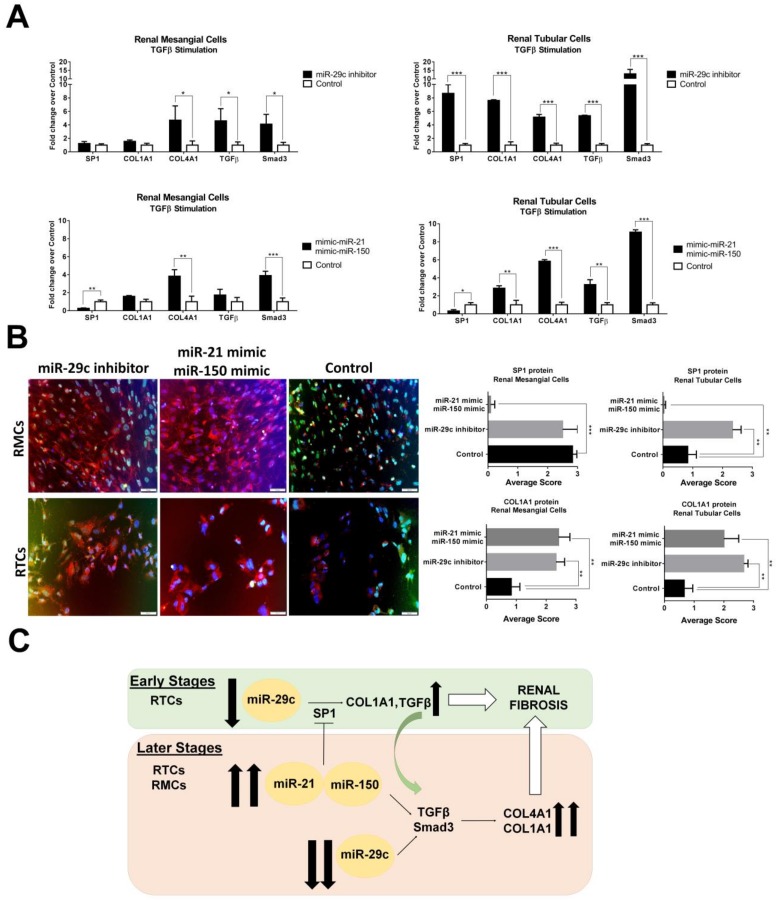
Overexpression of miR-21/miR-150 and inhibition of miR-29c increase profibrotic proteins. (**A**) Quantitative real-time RT-PCR analysis shows the relative mRNA levels of *SP1*, *COL1A1*, *COL4A1*, *TGFβ*, *Smad3* in TGFβ-stimulated renal mesangial cells (RMCs) and renal tubular epithelial cells (RTCs). Values were normalised using *GADPH* and fold change was calculated over control condition (mimic miR-control or anti miR-control). (**B**) Immunofluorescent staining exhibits COL1A1 expression in MRCs and TRCs with overexpression of miR-21/miR-150 and inhibition of miR-29c. SP1 staining was inhibited completely with the overexpression of miR-21/miR-150 in both type of cells. Scale bar = 50 µm. * *p* < 0.05, ** *p* < 0.005, and *** *p* < 0.0005. (**C**) Proposed mechanism for urinary exosomal miRNAs in early and later stages of LN renal fibrosis.

**Table 1 cells-08-00773-t001:** Clinical and laboratory characteristics of the study subjects.

Characteristics	Lupus Nephritis (n = 45)			Healthy Control (n = 20)
	Low CI (N = 18)	Moderate CI (N = 21)	High CI (N = 6)	
**Demographic**				
Age, year	33 ± 7	29 ± 5	30 ± 4	28 ± 5
Sex, male/female	7/11	8/13	3/3	8/12
Race/ethnicity, n (%)				
Caucasian	18	20	6	19
Hispanic	0	1	0	1
**Laboratory Parameters**				
Serum creatinine, mg/Dl	0.9 ± 0.3	1.1 ± 0.8	1.1 ± 0.6	0.7 ± 0.2
eGFR (mL/min)	92.3 ± 26.7	78.6 ± 29.3	78.8 ± 42.6	97.2 ± 31.7
BUN (mmol/l)	4.7 ± 1.2	4.6 ± 2.9	4.3 ± 1.7	5.0 ± 1.5
Anti-dsDNA Abs, IU/mL	333 ± 104	343 ± 107	357 ± 143	n.d.
Serum C3, mg/dL	69.5 ± 16.1	87.5 ± 12.1	74.5 ± 16.1	n.d.
Serum C4, mg/dL	8.3 ± 2.1	10.4 ± 3.5	11.2 ± 2.7	n.d.
Proteinuria, g/24 h	4.2 ± 2.4	3.6 ± 2.5	3.9 ± 4.7	n.d.
**Disease Index (SLEDAI-2K)**				
Total SLEDAI score	18 ± 2	14 ± 2	16 ± 3	n.d.
Renal Biopsy, n (%)				
Class, n (%)				
III	4	6	0	n.d.
IV	14	12	4	n.d.
V	0	3	1	n.d.
Activity Index	6.3 ± 3.7	5.3 ± 4.1	6.3 ± 3.2	n.d.
Chronicity Index	0.6 ± 0.4	3.5 ± 0.5	5.8 ± 1.2	n.d.

Values are means ± SE. BUN, blood urea nitrogen; eGFR, estimated glomerular filtration rate; anti-dsDNA, anti-double-stranded DNA; n.d., not determinate.

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
