# Peer review of "An Exosomal Urinary miRNA Signature for Early Diagnosis of Renal Fibrosis in Lupus Nephritis"

_cells, 2019, doi:10.3390/cells8080773_

Round 1

Reviewer 1 Report

The study of Solé et al. describes the use of urinary exosomal miRNAs for early diagnosis of Lupus nephritis (LN).

 urinary exosomal multimarker panel consisting of miR-21, miR-150 and miR-29c provide a non-invasive method to detect early renal fibrosis and predict disease progression in LN patients. In general, the study is well done. I have the following remarks:

Individual exosomal miRNAs have been described to be altered in urinary exosomes of LN patients, reducing novelty of this manuscript.

Longitudial analyses would be helpful in assessing the clinical value of urinaly exosomal miRNAs.

How do you explain that there were no correlations between the miRs and the classical clinical markers of kidney function? Did the CI group not correlate with Crea, BUN, eGFR? I would expect that.

I have a problem with the staining of COL1A1 and COL4A1. It should stain the extracellular collagens rather than the nuclei. I recommend checking this.

Can you speculate on the cells that release the exosomes? What might be the biological function of this?

Labeling in Fig. 4: “B” and “C” interchanged.

Author Response

Response to Reviewer 1 Comments

The Solé et al. study describes the use of urinary exosomal miRNAs for early diagnosis of Lupus nephritis (LN). A urinary exosomal multimarker panel consisting of miR-21, miR-150 and miR-29c provides a non-invasive method to detect early renal fibrosis and predict disease progression in LN patients. In general, the study is well done. I have the following remarks:

·         Individual exosomal miRNAs have been described to be altered in urinary exosomes of LN patients, reducing novelty of this manuscript.

The objective of the project is to study miRNAs related with lupus nephritis fibrosis in order to find a signature panel that could predict fibrosis formation. For this reason, we focus in miRNAs described in the literature. However, only miR-29c and miR-21 have been reported in urinary exosomes from LN patients [1,2]. MiR-150, miR-410 and miR-200-3p have not been described in urinary exosomes. MiR-150 and miR-410 have been described only in kidney tissue [3,4]. This is the first time a signature of miRNAs, as a multimarker biomarker panel, has been reported in the formation of LN renal fibrosis

[1] Solé, C.;Cortés-Hernández, J.; Felip, ML.; Vidal, M.; Ordi-Ros, J. MiR-29c in urinary exosomes as predictor of early renal fibrosis in lupus nephritis. Nephrol Dial Transplant, 2015, 30, 1488-96.

[2] Tangtanatakul P, Klinchanhom S, Sodsai P, Sutichet T, Promjeen C, Avihingsanon Y, Hirankarn N. Down-regulation of let-7a and miR-21 in urine exosomes from lupus nephritis patients during disease flare. Asian Pac J Allergy Immunol. 2018, 13, [Epub ahead of print].

[3] Zhou, H.; Hasni, SA.; Perez, P.; Tandom, M.; Jang, SI.; Zheng, C.; Kopp, JB.; Austin, H 3rd.; Balow, JE.; Alevizos, I.; et al. miR-150 promotes renal fibrosis in lupus nephritis by downregulating SOCS1. J Am Soc Nephrol, 2013, 24, 1073-87.

[4] Liu, D.; Zhang, N.; Zhang, J.; Zhao, H.; Wang, X. miR-410 suppresses the expression

of interleukin-6 as well as renal fibrosis in the pathogenesis of lupus nephritis. Clin Exp

Pharmacol Physiol, 2016, 43, 616-25.

·         Longitudinal analyses would be helpful in assessing the clinical value of urinary exosomal miRNAs.

In order to increase the quality of the paper, we have included a longitudinal study for the patients that have a repeat kidney biopsy. Second biopsies were only performed for lack of response to therapy, for this reason, we have only seven patients. In those, we have evaluated miRNA expression and correlated with renal chronicity index. We have included it in results (Page 6, lines 212-215) and in the Figure 2D.

·         How do you explain that there were no correlations between the miRs and the classical clinical markers of kidney function? Did the CI group not correlate with Crea, BUN, eGFR? I would expect that.

Proteinuria, creatinine, BUN and eGFR are considered the traditional biomarkers of prognosis and follow up but sometimes they do not reflect accurately the state of the kidney and renal biopsy continues to be the gold standard. Since miRNA expression seems to be an early marker of progression, we think that this is the reason that we did not observe correlation with Crea, BUN, proteinuria or eGFR. However, it would add great value to the current markers.

·         I have a problem with the staining of COL1A1 and COL4A1. It should stain the extracellular collagens rather than the nuclei. I recommend checking this.

You are right, it was a writing mistake. We have corrected it (Page 7, lines 233-234).

·         Can you speculate on the cells that release the exosomes? What might be the biological function of this?

We observed in LN kidney biopsies that protein targets of miRNAs (SP1 and collagens) are largely localized in tubular epithelial renal cells. So we speculated that the target cells of exosomes will be the renal tubular epithelial cells. They are probably a neglected mediator in renal fibrosis. Recent studies demonstrated that damage with a maladaptive repair of renal tubular epithelial cells leads to the progression of renal fibrosis [1, 2]. Their secretion of proinflammatory and profibrotic factors such as TGFβ1, CXCL1, IL-6 or IL-8 promotes renal fibrosis. In addition, renal tubular epithelial cells can also undergo epithelial-to-mesenchymal transition after injury, which loses some epithelial markers (E-cadherin) and acquisition of partial myofibroblast markers (SMA, vimentin, FSP-1) that are responsible for collagen synthesis and extracellular matrix deposition [3]. However, immune complex deposition in kidney produces secretion of inflammatory cytokines with an increase of mesangial proliferation and a progression to glomerulosclerosis. Glomerular injury produces high levels of angiotensin II, TGFB, CTGF, PAI-1 and NFKβ leading to immune response and extracellular matrix formation. An accumulation of them, contributes to renal fibrosis [4]. For this reason, in vitro experiments have been added in the manuscript focusing on the role of urinary exosomal miRNAs in mesangial renal cells and in tubular epithelial renal cells. We have included the results in a new section 3.6 (Page 7, lines 264-276) and in the discussion (Page 13, lines 351-368).

[1] Gewin, L. et al. TGF-β receptor deletion in the renal collecting system exacerbates fibrosis. J. Am. Soc. Nephrol. 2010, 21, 1334–1343.

[2] Liu, Y. Epithelial to mesenchymal transition in renal fibrogenesis: pathologic significance, molecular mechanism, and therapeutic intervention. J. Am. Soc. Nephrol. 2004, 15, 1–12.

[3] Nogueira A, Pires MJ, Oliveira PA. Pathophysiological Mechanisms of Renal Fibrosis: a review of animal models and therapeutic strategies. In vivo. 2017, 31, 1-22.

[4] Meng XM, Nikolic-Paterson DJ, Lan HY. Inflammatory processes in renal fibrosis. Nat Rev Nephrol. 2014, 10, 493-503.

·         Labeling in Fig. 4: “B” and “C” interchanged.

It has been corrected in the reviewed Figure 3.

Reviewer 2 Report

Manuscript Number: cells-526185

The manuscript titled “An exosomal urinary miRNA signature for early diagnosis of renal fibrosis in Lupus Nephritis” by Solé C., et al., was aimed to identify the exosomal urinary microRNAs that differentially expressed in LN patients, and explored the putative biological targets affected that may contribute to the pathogenesis of LN. From their results the authors suggest that the expressions of miR-21, miR-150 and miR-29c may be useful for diagnostic and prognostic markers for LN.

This is a well written manuscript with high originality. However, the authors did not provide sufficient evidences to support the regulatory effect of miRs on the expressions of target genes in respective cellular model. The Reviewer feels results from in vitro study might be essential for the authors to claim the effects of these dysregulated miRs on the fibrosis, and would be better fit with the scope of this journal. Based on the content of current manuscript, the Reviewer regrets to reject the manuscript for publication in present form.

The authors may consider to further improve their manuscript by addressing following concerns,

Materials and Methods

1.       2.1 Information for IRB approval is essential for using patients’ specimen. The authors may wish to state relevant information in the manuscript.

2.       2.3 The amounts of exosome and exosomal RNAs extracted from 50 ml urine.

3.       The completed list of the miRNA-seq raw data should be deposited to GEO before acceptance of manuscript and the link should be provided in revised manuscript.

4.       The analyzed miRNA-seq data should be provided as supplemental data and the authors should address further on the selection of 5 microRNAs for downstream analyses.

Result

5.       Figure 1A; The density of exosome is low, the reviewer only could find one exosome like body. The authors may wish to explain the reason or replace the photo with clear labels.

6.       Figure 1B; The authors did not provide detailed compositions/list of miRs in exosome and cellular pellet. And the authors may wish to define the term of “specific miRNAs” (line 183)

7.       The authors should perform miR/3’UTR luciferase assay and gain/loss-of-function assay to demonstrate the regulatory effects of miRs on their respective target genes.

8.       The authors may wish to manipulate the levels of miRs and examine the fibrosis in cells to further support their working hypothesis.

Author Response

Response to Reviewer 2 Comments

The manuscript titled “An exosomal urinary miRNA signature for early diagnosis of renal fibrosis in Lupus Nephritis” by Solé C., et al., was aimed to identify the exosomal urinary microRNAs that differentially expressed in LN patients, and explored the putative biological targets affected that may contribute to the pathogenesis of LN. From their results the authors suggest that the expressions of miR-21, miR-150 and miR-29c may be useful for diagnostic and prognostic markers for LN.

 This is a well written manuscript with high originality. However, the authors did not provide sufficient evidences to support the regulatory effect of miRs on the expressions of target genes in respective cellular model. The Reviewer feels results from in vitro study might be essential for the authors to claim the effects of these dysregulated miRs on the fibrosis, and would be better fit with the scope of this journal. Based on the content of current manuscript, the Reviewer regrets to reject the manuscript for publication in present form.

 The authors may consider to further improve their manuscript by addressing following concerns,

Materials and Methods

1.        Information for IRB approval is essential for using patients’ specimen. The authors may wish to state relevant information in the manuscript.

All the studies performed in Vall Hebron Institute Research have been evaluated by Vall Hebron Ethics Committee. We have sent the evaluation to the editor. We have included this information in the manuscript (Page 3, lines 78-79).

2.        The amounts of exosome and exosomal RNAs extracted from 50 ml urine.

Using FluorCet Exosome Quantitation Kit (SBI), we have quantified the amount of exosome extracted from 50 mL urine. The amount obtained were similar in all groups. We have included it in the manuscript (Page 3, line 102-104 and Page 5 line 167-168).

3.       The completed list of the miRNA-seq raw data should be deposited to GEO before acceptance of manuscript and the link should be provided in revised manuscript.  The analyzed miRNA-seq data should be provided as supplemental data and the authors should address further on the selection of 5 microRNAs for downstream analyses.

We have submitted the miRNA-seq in GEO database, but for the small number of samples, it was not accepted. The objective of miRNA-seq in our study was not a screening to know which miRNAs are related in LN fibrosis formation. We performed it using the urine from three active LN patients and we compared which is the best source to study the miRNAs (urine or urinary exosomes). However, we did not compare the results between lupus nephritis CI groups. The five miRNAs selected in the study are those described in the literature, not from the miRNA-seq results. We performed a qPCR-RT comparative study only with the five miRNAs selected using urine and urinary exosome (Page 5, lines 175-178) and we specified better the objective of the paper in order to avoid confusion, (Page 2, lines 71-74). We only mention the miRNA-seq in the manuscript but we have included all the information in Supporting Information (Figure S4).  

Result

5.       Figure 1A; The density of exosome is low, the reviewer only could find one exosome like body. The authors may wish to explain the reason or replace the photo with clear labels.

We have replaced the picture for one more representative (Figure 1A).

6.       Figure 1B; The authors did not provide detailed compositions/list of miRs in exosome and cellular pellet. And the authors may wish to define the term of “specific miRNAs” (line 183)

MiRNA-seq was performed only to compare the source of urinary biomarkers. It was not done to find differentially expressed miRNAs between lupus nephritis groups. For this reason, we have not provided a detailed list of them. In order to clarify, we have modified it in the manuscript (Page 5, lines 175-178) explaining better the role of the miRNA-seq analysis. In addition, we have included all the information about miRNA-seq (methodology, yield/size library, the distribution analysis) in the supporting information (Methodology and Figure S4).

7.       The authors should perform miR/3’UTR luciferase assay and gain/loss-of-function assay to demonstrate the regulatory effects of miRs on their respective target genes.

We have done these assays to demonstrate the regulatory effects of miR-21, miR-150 and miR-29c to SP1 target. All these results have been included (Page 7, lines 235-242).

8.       The authors may wish to manipulate the levels of miRs and examine the fibrosis in cells to further support their working hypothesis.

We have performed transfection of miRNAs in order to study their role in LN renal fibrosis. We have included a new section in the results (3.6, Page 7 lines 243-256), a new figure (Figure 4) and we discuss the results obtained (Page 13, lines 351-373).  

Round 2

Reviewer 1 Report

The manuscript looks fine now. Check line 480 "profibrotic proteins".

Reviewer 2 Report

The authors have attended the concerns from the Reviewer. No further inquiries raised.